global mental health delivery; HIV; interventions; community engagement; peer support

**Corresponding author:**
Caroline Wang Kokubun;
Email: caroline.kokubun@emory.edu

# A global scoping review of task shifting and sharing interventions to improve the mental health of people living with HIV/AIDS

Caroline Wang Kokubun[1] [ID], Madelyn Smith Carlson[1], Benjamin George Druss[1,2], Briana Ashley Woods-Jaeger[1], Ameeta Shivdas Kalokhe[3,4] and Jessica McDermott Sales[1]

[1]Department of Behavioral, Social, and Health Education Sciences, Rollins School of Public Health, Emory University, Atlanta, GA, USA; [2]Department of Health Policy and Management, Rollins School of Public Health, Emory University, Atlanta, GA, USA; [3]Hubert Department of Global Health, Rollins School of Public Health, Emory University, Atlanta, GA, USA and [4]Division of Infectious Diseases, Emory School of Medicine, Emory University, Atlanta, GA, USA

## Abstract

People living with HIV/AIDS (PLWH) often experience co-morbid/co-occurring mental health conditions, e.g., depression, anxiety, and post-traumatic stress disorder (PTSD). In resource-limited settings, where provider shortages are common, task shifting and task sharing (i.e., service delivery by non-professionals) are recommended strategies to promote access to and utilization of mental health and psychosocial support (MHPSS) services among PLWH. We conducted a global scoping review of the literature on MHPSS task shifting and sharing intervention studies for PLWH. Data extracted and summarized included study characteristics, intervention components, whether trauma informed study design, how lay health workers (LHWs) were identified and trained to deliver MHPSS services, and findings related to mental health outcomes. Results indicated that from 2013 through 2022, published intervention research concerning task shifting and sharing approaches was much more prolific in low- and middle-income countries than in high-income countries. MHPSS interventions delivered by a variety of LHWs yielded promising associations on an array of mental health outcomes, including PTSD/trauma and suicidality, though understudied. Underreported details regarding LHW recruitment/selection, compensation, supervision and assessment made it difficult to identify common or best practices. Further research is needed to facilitate the adoption and implementation of MHPSS task shifting and sharing interventions.

## Impact statement

In low- and middle-income (LMIC) countries, as well as resource-limited settings in high-income countries (HICs), implementing task shifting and task sharing interventions may increase the capacity of healthcare organizations to provide mental health and psychosocial support (MHPSS) services to people living with HIV/AIDS (PLWH). Training lay health workers (LHWs), such as non-specialist health care workers, community health workers, and peers to deliver MHPSS services can improve access to and engagement in mental health care among PLWH and may provide relief to overburdened health care systems. This global scoping review describes and characterizes the extent to which LHWs have been mobilized to deliver MHPSS services to PLWH. Specifically, it summarizes the literature on this topic over a 10-year period (2013–2022), describing the characteristics of interventions studied, results, implementation barriers and facilitators, study limitations, gaps in knowledge, and areas for future research.

Findings from this review indicate that task shifting and sharing are widely used strategies among LMICs and have been relatively successful at improving mental health outcomes among PLWH. Researchers and practitioners can use this scoping review as a resource to guide the future development of MHPSS task shifting and sharing studies and interventions for PLWH.

## Introduction

In 2023, the World Health Organization (WHO) estimated 39.9 million people were living with HIV worldwide, including 1.3 million (3.26%) newly diagnosed; that same year, an estimated 630,000 people died of HIV-related causes globally (WHO, 2024b). Low- and middle-income countries (LMICs) shoulder a disproportionate share (over 80% in 2020) of the global burden of HIV (Allel et al., 2022). African countries are most affected (Shao and Williamson, 2012; WHO,

2024b), comprising 26 million (65.16%) cases, 640,000 (49.23%) new infections, and 61.90% of HIV-related deaths globally among people living with HIV (PLWH) (WHO, 2024b).

PLWH experience disproportionately high rates of mental health conditions, e.g., depression, anxiety, and post-traumatic stress disorder (PTSD) (LeGrand et al., 2015). The consequences of not addressing the mental health concerns of PLWH are reduced retention in care (Anderson et al., 2019) and adherence to antiretroviral therapy (ART) (Crepaz et al., 2008; Boarts et al., 2009; Pence, 2009; Mayston et al., 2012; LeGrand et al., 2015; Anderson et al., 2019; Smith and Cook, 2019; Hou et al., 2020), disease progression (e.g., increased viral loads [Anderson et al., 2019; LeGrand et al., 2015], lower CD4 cell counts [Anderson et al., 2019; LeGrand et al., 2015; Pence, 2009], and virologic and treatment failure [LeGrand et al., 2015; Pence, 2009]), poorer quality of life (QoL) (Crepaz et al., 2008; Degroote et al., 2014; Penner-Goeke et al., 2015; Kabunga et al., 2024), and increased mortality (Pence, 2009; Anderson et al., 2019). Interventions addressing trauma among PLWH are needed, as they are few and understudied (LeGrand et al., 2015).

Evidence from mental health and HIV intervention literature supports the implementation of task shifting and task sharing to deliver mental health and psychosocial support (MHPSS) interventions to PLWH. The WHO developed task shifting and sharing as strategies to alleviate the burden on health care providers (e.g., mental health professionals), which are often used in LMICs to improve health care access despite limited resources, e.g., training lay health workers (LHWs), such as non-specialist health care workers (HCWs; e.g., nurses and adherence counselors), community health workers (CHWs), and peers (i.e., fellow patients) to deliver needed services (Padmanathan, 2013; Javanparast et al., 2018).

LHWs work in paid/compensated positions or as volunteers, occupying a broad spectrum of roles to improve community health and competence by providing information, practical assistance, and social support (Eng and Parker, 2002; Scott, 2009). They are often "natural helpers" who have shared or lived experiences and can assist healthcare organizations in several ways by improving: (1) health practices by increasing knowledge of and access to health resources and by facilitating the use of services; (2) institutional awareness and responsiveness to community needs; and (3) coordination of services through collaborative relationships with healthcare providers (Eng and Parker, 2002).

Although substantial evidence exists of the effectiveness of LHW MHPSS interventions in LMICs (Chibanda et al., 2015), little is known of their impact in high-income countries (HICs) (Sikkema et al., 2015; Javanparast et al., 2018). In recent years, implementation research concerning the mobilization of LHWs (particularly CHWs) to strengthen the HIV care continuum has become a priority of the United States (U.S.) National Institutes of Health (NIH) (NIH, 2020a, 2020b, 2020c). Han et al., (2018) obtained mixed results when conducting a systematic review of 13 CHW interventions, seven of which were conducted in the U.S., to improve psychosocial health outcomes in PLWH. Although none of the interventions yielded statistically significant improvements in mental health outcomes, most studies examined had various methodological and design flaws that may have biased results (e.g., psychosocial outcomes were not primary outcomes, sample size calculations were not performed in advance, a priori power analyses were not conducted for psychosocial variables, unstandardized psychological outcome measures were used, studies had high attrition, and follow-up was short-term) (Han et al., 2018). Systematic reviews (Boucher et al., 2020; Berg et al., 2021; Øgård-Repål et al., 2021) of peer interventions for PLWH yielded more promising

results: three (Simoni et al., 2007; Brashers et al., 2017; Cunningham et al., 2018) of four U.S. studies (Simoni et al., 2007; Brashers et al., 2017; Cunningham et al., 2018; Merlin et al., 2019) reported significant improvements in mental health outcomes, including reductions in depressive symptomology/decline in depression and increased use of mental health care.

Studies indicate that peer-delivered interventions are both acceptable and feasible for people in need of mental health services (Repper and Carter, 2011; Miyamoto and Sono, 2012; Padmanathan, 2013), as well as PLWH (Wewers et al., 2000; Wolitski et al., 2005; Purcell et al., 2007; Webel, 2010; Horvath et al., 2013; Enriquez et al., 2014; Steward et al., 2018; Merlin et al., 2019; Boucher et al., 2020). Furthermore, the WHO champions the mobilization of LHWs, especially peers, as an acceptable and even preferred means of delivering services to PLWH (WHO, 2016; Haberer et al., 2017; Boucher et al., 2020). In LMICs, peer support, mentorship, and counseling interventions have been effective in reducing symptoms of depression and anxiety among PLWH (Mayston et al., 2012; Sikkema et al., 2015; Asrat et al., 2020). For PLWH experiencing mental health issues, peers may be mobilized to greater effect than other LHWs due to their lived experience and ability to better relate to their clients' concerns, serving as care navigators and role models for recovery (Berg et al., 2021; Øgård-Repål et al., 2021; Krulic et al., 2022). Understanding how different LHWs compare in effectiveness to one another and usual care may influence LHW mobilization for future delivery of MHPSS services.

In some countries, LHWs have been empowered to deliver MHPSS services to PLWH, including therapeutic interventions, such as cognitive behavioral therapy (CBT) (Chibanda et al., 2015; Asrat et al., 2020). However, it is important to note that practice requirements may vary by type of therapy and across countries and implementing institutions/organizations (e.g., licensing/certification, advanced degree/higher education, and specialized training), posing potential barriers to service delivery. Improving awareness of which MHPSS interventions are being delivered to PLWH, by whom (including their qualifications and training), where, and to what outcome will advance our understanding of how LHWs can help close the mental health treatment gap and reinforce steps along the HIV care continuum.

Past reviews have extensively covered task shifting and sharing interventions for PLWH but have not focused on mental health outcomes or have been more limited in scope (e.g., only focused on one type of intervention) or setting (e.g., LMICs only). We were interested in how LHWs have been mobilized in different contexts and settings to deliver MHPSS services, identifying current trends and gaps in knowledge, and informing future research. To this purpose, we conducted a global scoping review (inclusive of both LMICs and HICs) to characterize the recent literature (publications from 2013 through 2022) on MHPSS task shifting and sharing interventions for PLWH.

Additionally, due to the scarcity of empirically supported trauma interventions concerning PLWH identified by LeGrand et al. (2015), as a secondary interest, we sought to identify if and how trauma influenced intervention design and selection among studies reviewed, and whether a gap remains.

## Methods

This paper is guided by the preferred reporting items for systematic review and meta-analyses (PRISMA) extension for scoping reviews (PRISMA-ScR) checklist and explanatory paper (Tricco et al., 2018).

### Eligibility criteria

To be included in this review, intervention studies met the following eligibility criteria: (1) they were published in a peer-reviewed journal from 2013 through 2022, (2) they were written in English, (3) the study population was adult PLWH aged 18+, (4) the primary/main study outcome(s) were related to improving mental health, and (5) a task shifting or task sharing approach was used to deliver MHPSS services. Quantitative and qualitative study designs were eligible for inclusion, provided they met all other criteria; however, literature reviews were not. Additionally, papers were excluded if the intervention studied was not fully implemented at the time of publication. Studies targeting youth and adolescents that included some participants aged 18+ were also excluded.

We used a broad definition for mental health outcome, which included scores from the measurement/assessment of a specific disorder (e.g., depression, anxiety, and PTSD) and related constructs (e.g., QoL and resiliency [Chuang et al., 2023; González-Blanch et al., 2018; Hu et al., 2015]), as well as changes in access to and engagement in mental health services (e.g., linkage to services and a number of mental health visits). For mental health to be considered a primary study outcome, it had to be explicitly stated as the main purpose for the conducted research. If mental health was one of several outcomes but not prioritized, it was not considered a primary outcome.

Pilot studies assessing acceptability and feasibility were only included if improving mental health was the intervention's primary purpose, and mental health outcomes were also reported. Task shifting and sharing were defined as intervention strategies delegated to or collaboratively mobilizing non-specialist HCWs and other LHWs (e.g., CHWs and peers) to provide services to PLWH. Interventions delivered by research staff or individuals with prior education or training, such as a degree or certificate in mental health, were excluded (neither task shifting nor task sharing).

### Search strategy and selection process

An initial search of the PubMed/MEDLINE database was conducted to explore results from a combination of key search components and variations of related terms: i.e., (population, e.g., PLWH) AND (mental health, e.g., depression, anxiety, and PTSD) AND (intervention/program) AND (task shifting/sharing, e.g., lay health). When the study team was satisfied with the relevance of the search returns, search terms and conditions were finalized and translated to other databases (i.e., PsycINFO, Global Health, Web of Science, and SCOPUS) to source additional studies (File S1). Filters were also applied to limit search results to English-language (e.g., "Eng[lang]") publications from 2013 through 2022 (e.g., 2013/01/01 "[PDAT]":2023/01/01 "[PDAT]").

All studies retrieved from these searches were uploaded to Covidence, an online systematic review management platform. The software identified and flagged duplicates for review, which the principal investigator (PI) removed where applicable. The articles were then screened for selection and data extraction. The PI conducted the initial title and abstract review of each study retrieved. Although literature reviews in our search were not included, their reference lists were examined to ensure that relevant studies were still captured.

Two reviewers (the PI and another doctoral student) then read the full text of the remaining articles and independently decided whether each study met the criteria for inclusion. Coding categories for study exclusion criteria were created in Covidence, allowing reviewers to provide their rationale for excluding studies. Although studies may have had multiple reasons for exclusion, Covidence software permitted only one rationale selection. Thus, reviewers selected their primary reason for exclusion, although this was not systematically done. Reviewers then met to discuss and resolve discrepancies in study selection flagged by Covidence. When they disagreed, a third reviewer (a subject matter expert) was consulted and a consensus was reached before a final decision was made to include or exclude the study. Study quality was not assessed and did not influence selection.

### Data extraction

After the studies were selected, two reviewers extracted relevant data from a close reading of each publication and entered it into a Microsoft Excel spreadsheet table. Reviewers pulled the following information from each study: (1) publication year; (2) study design and comparison groups; (3) location and setting; (4) data collection period; (5) study aim; (6) sample size and demographic data; (7) intervention characteristics and whether trauma was a consideration in the design or selection of an intervention; (8) LHW characteristics, roles, training, compensation, and intervention responsibilities; and (9) mental health results. Barriers and facilitators to implementation were documented and study limitations were noted to identify gaps for further research.

### Data analysis

Data were explored through close reading, paraphrasing, and reorganizing and restructuring tables in Microsoft Excel. The study team then generated summaries for each data item and identified themes and patterns that emerged across studies. Summary tables were created in Microsoft Word to visually represent findings from this review and are presented in the next section.

## Results

### Study selection

Our literature search retrieved 1,504 papers; after removing duplicates, 885 remained. Although literature reviews were excluded, we identified 14 additional papers from reference lists of relevant reviews among our search returns and screened a total of 899 titles and abstracts. After reviewing titles and abstracts, we read the full text of 104 papers to assess their eligibility and excluded 75. Ultimately, 29 papers were included in our review, two of which reported findings from the same intervention and were considered together as one study ($N = 28$). See Figure 1 for details of the selection process.

### Study characteristics

Papers included in our review were published (based on issue/citation date) from 2013 through 2022. The most productive year was 2014 with six papers. However, the second half of the review period accounted for 58.6% of the overall sample, and an increase in productivity in recent years (2020–2022) accounted for 37.9% of papers. Of the 28 papers that reported when data collection took place, recruitment often coincided with baseline data collection, and the average total duration of data collection was 15.6 months

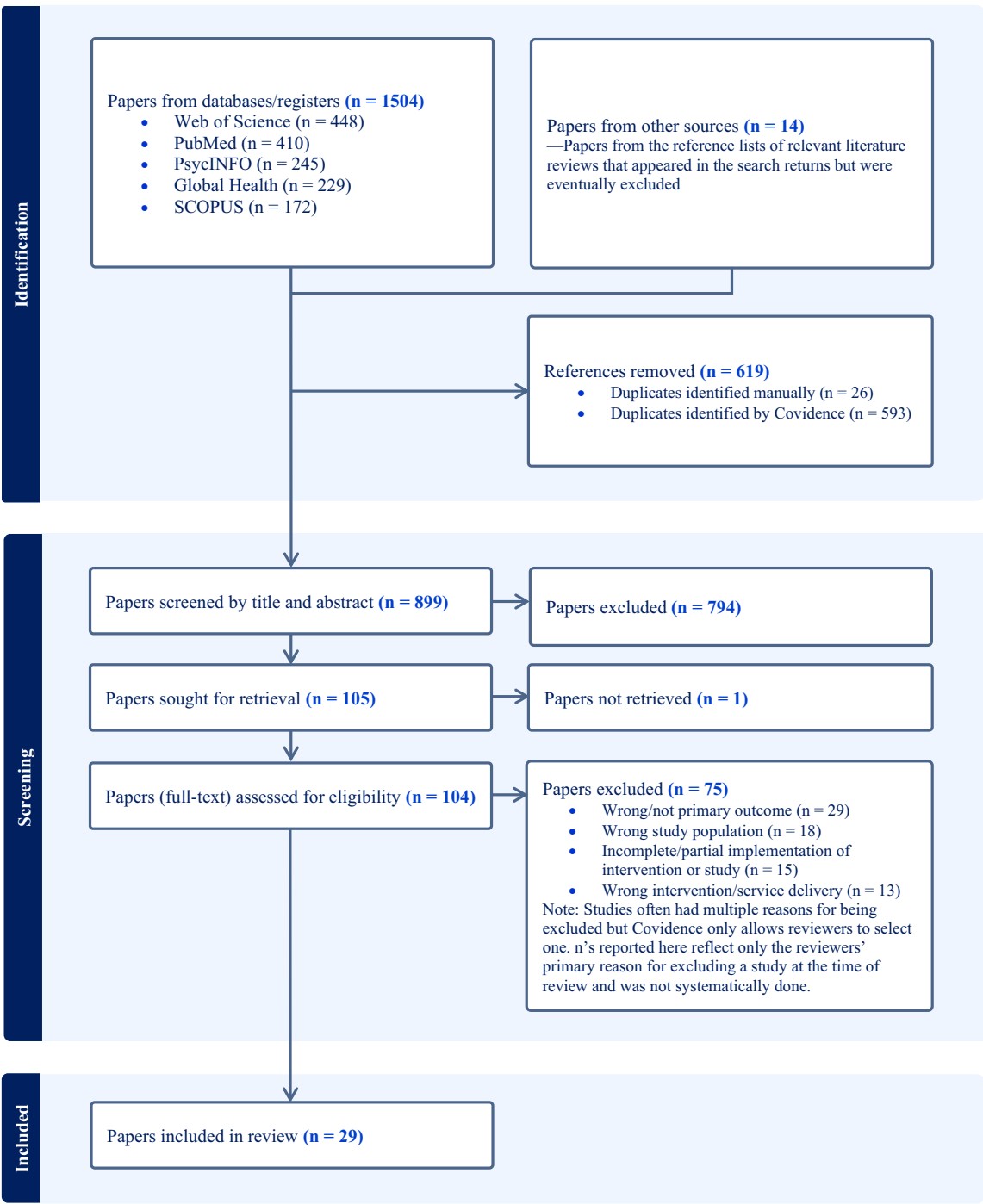

**Figure 1.** PRISMA flow diagram of papers reviewed (N = 1,518).

(range: 2–38). The characteristics of individual studies are presented in Table 1.

### Study design

Half of the studies in this review (*n* = 14) were randomized controlled trials (RCTs), including cluster randomized designs (cRCT; *n* = 4). Other designs used included non-randomized, single-arm, cross-sectional, prospective cohort, case series, times series, and pre-/post-tests. Most studies elected to use quantitative research methods, however, some used qualitative and mixed methods.

Comparators were baseline and follow-up assessment(s), usual care, enhanced usual care, a limited version of the intervention, or an active control. Studies did not compare nor assess differences in intervention efficacy/effectiveness based on LHW classification. However, Wagner et al., 2016 compared results between two non-specialist HCW intervention models – "structured protocol" via trained nurses and "clinical acumen" via primary care providers (mostly nurses) – rather than using a control. Additionally, Myers

**Table 1.** Characteristics of included studies (N = 28)

| Citation | Study design | Intervention description | Location/setting | Sample size/eligibility | Lay health workers | Mental health outcome(s) | Results |
|---|---|---|---|---|---|---|---|
| (Abas et al., 2018) | Individualized RCT where the experimental intervention was compared to enhanced usual care | Trained and supervised lay adherence counselors providing six sessions of PST for depression and adherence in a stepped-care model | University-affiliated government clinic providing comprehensive HIV care in Zimbabwe | 32 adults (18+) on ART for at least four months with mild depression that are at risk for poor adherence | Peers | Depression: PHQ–9 and SSQ–14 | PHQ–9: Statistically and clinically significant reduction in average depression score from moderate to no depression and when compared to mild depression in control group at follow-up; SSQ–14: reduction in mean score but the effect was not significant |
| (Andersen et al., 2018) | Non-controlled interrupted time series | Nurse-delivered CBT intervention for depression and adherence (CBT-AD) | Two HIV primary care clinics in Khayelitsha, Western Cape, South Africa | 14 adults (18+), seven from each site, with MDD and using ART (must have been diagnosed with HIV at least six months prior) | Nurses | Depression: self-rated (CES-D) and clinician-rated (Hamilton Depression Rating Scale: HAM-D)* | Very large and statistically significant reductions in mean score on CES-D and HAM-D |
| (Asrat et al., 2021) | Single-arm intervention mixed methods study with pre- and post-assessments | One session of pre-group individual counseling and 8 weekly sessions of peer-administered group IPT (5–10 participants per group) for treatment of depressive symptoms | Referral hospital in northwest Ethiopia | 31 adults (18+) using ART with MDD | Adherence supporters and case managers who are also HIV+ (peer counselors) and nurses (supervisors) | Depression (PHQ–9) and QoL (WHOQOL-HIV-BREF-Ethiopia)** | Statistically significant reduction in mean score for PHQ–9 and statistically significant increase in mean score for WHOQOL-HIV-BREF-Ethiopia; in-depth interviews: participants reported that the intervention reduced depressive symptoms and peer counselors gave them hope |
| (Chen et al., 2018) | RCT | Self and family management intervention (SAFMI): three intervention sessions consisting of biofeedback for relaxation, family support, anxiety, stress, depression management, cognitive-behavioral management skills, and psycho-education to decrease depressive symptomology | Hospital (Beijing) and public health clinic (Shanghai), China | 82: 41 women living with HIV (18+) who had disclosed status to at least one family member and 41 family members | Nurses | Depression (CES-D) | Statistically significant lower probability and odds of clinically meaningful depressive symptomatology scores at three months than women in the TAU group after controlling for their respective baseline values |
| (Davis et al., 2021) | Non-controlled pilot study (baseline and 12-month follow-up) | Health navigation to promote linkage to care and an emotional well-being intervention for newly diagnosed/re-engaged patients | Hospital infectious disease clinic in Guatemala City, Guatemala | 346 MSM (18+) receiving care at the site | CHWs and peers | Depression (PHQ–2), anxiety (General Anxiety Disorder–2: GAD–2), and level of emotional support (e.g., navigator interactions related to mental health) | Statistically significant increase in navigator–participant interactions; high levels of informational navigation support were significantly associated with improved anxiety but more frequent in-person navigation was significantly associated with worsened anxiety; navigation was not associated with depression |
| (Duffy et al., 2017) | Multi-phase pilot study using a mixed-method exploratory sequential | Stepped care model: screening for common mental disorders; basic | Nine Zimbabwe primary care clinics (five urban and four rural) in the initial | PLWH receiving care from study sites (Phase 2: | CHWs, nurses, and traditional | Depression and anxiety screenings (SSQ and ACS) and referrals | There were 159 (23%) positive screens for depression and/or anxiety in Phase 2 and 88 (28%) positive screens in Phase 3; suicidal |

(Continued)

| Citation | Study design | Intervention description | Location/setting | Sample size/ eligibility | Lay health workers | Mental health outcome(s) | Results |
|---|---|---|---|---|---|---|---|
| | design that employed a phased descriptive approach with a longitudinal cohort | counseling and therapeutic interventions for positive screens; referrals to higher-level mental health services (i.e., medication, counseling therapy, and psychosocial services); and protocol development for mental health emergencies (e.g., suicidal ideation) | pilot (Phase 2); five of the highest performing clinics (four urban and one rural) in the refined pilot (Phase 3) | nine clinics and Phase 3: five clinics). SSQ: Phase 2: 703 and Phase 3: 312 Abbreviated Community Screen (ACS): Phase 2: 182 and Phase 3: 123 | medicine practitioners | | ideation was identified in 54 (61%) individuals among positive screens in Phase 3; there were 58 (32%) positive community screens for "sad" or "worry" questions in Phase 2 and 66 (54%) in Phase 3; SSQ nurse-initiated referrals increased from 136 (86%) to 154 (175%) in Phase 3, although community-level referral rate declined from 54 (93%) to 55 (83%) |
| (Fajriyah et al., 2018) | Cross-sectional | Solo Plus peer support group | Hospital in Surakarta, Central Java, Indonesia | 100 Solo Plus HIV peer support group participants | Peers | Depression and QoL (scales not reported) | Statistically significant increase in QoL of PLWH with involvement in peer support group; QoL significantly decreased with stigma and depression |
| (Garfin et al., 2019) | Baseline and 6-month follow-up as part of an RCT using a factorial design | Accredited Social Work Activist (ASHA) intervention: CHW accompaniment to group educational sessions, weekly check-ins, medical accompaniment, and logistical/medical/social support | Community health centers and primary care centers near four high-HIV prevalence sites in rural Andhra Pradesh, India | 600 women (ages 18–50) with young children (ages 3–8) that had been on ART for at least three months with CD4 levels ≥100 cells/mm3 | CHWs | Depression (CES-D) and QoL (10-item Quality of Life Enjoyment and Satisfaction Questionnaire)*** | Three classes identified: 1) highest social resources/lowest depression (highest QoL at baseline), 2) some social resources/ highest depression, and 3) lowest social resources/higher depression (lowest QoL at baseline); at six-month follow-up, all groups reported significant improvements to QoL and depression; depression is an important predictor of QoL |
| (Pence et al., 2014; Gaynes et al., 2015) | Pilot non-controlled before and after (baseline and 4 months follow-up) | Adaptation of measurement-based care (MBC) for depression management to measure symptoms and treatment response, assess tolerability, and provide decision support to non-psychiatric prescribers | Hospital/AIDS treatment center in Bamenda, Cameroon | 55 PLWH (aged 18–65) with MDD | Nurses, social workers, and other HCWs; nurse/ pharmacy attendant | Depressive severity (PHQ–9), adaptive and maladaptive coping styles (Brief COPE), self-efficacy (Watt Self-Efficacy Scale), and suicidal ideation | There was a significant decrease in PHQ–9 from baseline to one-month follow-up and 90% of participants achieved a PHQ–9 score below 5 (remission); sustained reductions in suicidal ideation |
| (Guy et al., 2022) | Secondary analysis of a quasi-experimental community-based participatory research (CBPR) study (non-controlled before and after) | Four weekly peer-led group sessions for the Prepare2Thrive peer support intervention | Hospital-based Ryan White-funded HIV clinic in Chicago, IL, USA | 18 African American adults (18+) with symptoms of mental illness | Peers | Cognitive escape (Cognitive Escape Scale) | Significant indirect effect of HIV-related discrimination on cognitive escape coping through internalized stigma pre-intervention (partial mediation); negative association between HIV discrimination and cognitive escape coping was no longer significant and there was no evidence of mediation post-intervention |
| (Han et al., 2020) | Randomized controlled pilot | CBT and relaxation training in 10 weekly group-based sessions (two hours/session) that was adapted from a simplified cognitive behavior stress management/expression-supportive (CBSM+) framework to provide participants with psychological knowledge and skills | Public health center in an infectious disease treatment hospital in Shanghai, China | 21 adults (18+) receiving ART that had a PHQ–9 score of 2+ and were not seeing a psychologist/ psychiatrist | Nurse and volunteer assistants (social workers) | Depression and anxiety (PHQ–4, GAD–2, and Hospital Anxiety and Depression Scale) | Significant difference in anxiety scores with greater improvement in the intervention group; depression scores did not differ significantly |

| Citation | Study design | Intervention description | Location/setting | Sample size/ eligibility | Lay health workers | Mental health outcome(s) | Results |
|---|---|---|---|---|---|---|---|
| (Kaaya et al., 2022) | Pair-matched cRCT | Healthy Options intervention: PST and CBT components implemented by trained peer facilitators in a stepped care model over 8 weekly small group sessions | 16 sites (government-managed or NGO-supported clinics and satellite dispensaries providing integrated prevention of mother-to-child transmission of HIV services for ≥350 HIV-positive pregnant women) in Dar es Salaam, Tanzania | 742 pregnant women (18+) at gestational stage <30 weeks on ART with MDD planning to receive postpartum care at site (395 in intervention clinics, 347 in control clinics) | CHWs | Depression (PHQ–9) | The risk of depressive symptoms consistent with MDD in women in intervention clusters was significantly lower than the risk in women in control clusters at nine months postpartum and the risk in control clusters at six weeks postpartum; the PHQ–9 score was significantly lower in women in the intervention group compared to the control group at nine months postpartum and at six weeks postpartum |
| (Li et al., 2021) | Three-arm RCT: Three Good Things (TGT) intervention vs. TGT with electronic social networking (TGT-SN) vs. control (weekly emails containing information on promoting mental health) | TGT-SN groups (11–30 people in each) involving daily engagement (i.e., posting three positive experiences, reading other members' posts, and giving feedback with comments and likes) | Local NGO (one of the largest national LGBT+ organizations) in China | 415 MSM (18+) who were regular users of the QQ social network and who had anal sex within the last six months, had been diagnosed with HIV for at least three months with no severe medical conditions, and were not using psychiatric/ psychological counseling services | Peers | Depression (CES-D) and anxiety (GAD–7) | The main effects of TGT-SN in reducing depression were statistically non-significant; the participants of the TGT-SN group showed significantly lower anxiety symptoms and negative affect over time compared with those of the control group; no significant main effect was found for any secondary outcomes for the TGT-only group |
| (Li et al., 2017) | Non-randomized community trial without parallel control | Love Home social support and care model: small groups led by a nurse, social volunteer, and peer volunteer for health education, stress management, social interaction, and therapy support | Outpatient HIV infection center in a hospital in Beijing, China | 494 adult PLWH (18+) | Nurses and peer volunteers (for psychological interventions and therapy support) | QoL (36-item Medical Outcomes Study Short Form) | Significant differences in QoL, including mental health |
| (Masquillier et al., 2014) | RCT that is a component of a prospective cohort study | Bi-weekly peer adherence support for 18 months, as well as nutrition support for a subset of participants | 12 public ART clinics in Lejweleputswa, Motheo, Thabo Mofutsanyana, Fezile Dabi, and Xhariep, South Africa | 653 PLWH living in town/village with a study site | Peers | Hope (Adult State of Hope Scale) | Peer adherence support had no direct effect on level of hope; significant positive interaction between family functioning and peer adherence support at second follow-up (better family functioning increases positive effect of peer adherence support on hope) |
| (Meffert et al., 2021) | RCT: effectiveness-implementation hybrid type I trial | 12 weekly one-hour sessions of IPT | Family AIDS care, education, and services (FACES) HIV care and | 256 women living with HIV | CHWs | Major depressive disorder (Beck Depression Scale: BDI-II) and PTSD | Participants who received IPT had significantly lower odds of MDD after intervention, lower odds of PTSD, and lower |

**Table 1.** (*Continued*)

| Citation | Study design | Intervention description | Location/setting | Sample size/ eligibility | Lay health workers | Mental health outcome(s) | Results |
|---|---|---|---|---|---|---|---|
| | | | clinical research clinic in Kisumu, Kenya | | | (Posttraumatic Stress Disorder Checklist-Civilian) | odds of MDD-PTSD; significant group differences in depressive and PTSD symptoms were observed at three months (when groups differed in receipt of IPT); gains were maintained at follow-up visits |
| (Myers et al., 2022) | Three-arm cRCT: dedicated care vs. designated care vs. treatment as usual | MIND Program: three sessions of an evidence-based psychological intervention based on motivational interviewing and PST with optional booster session | 24 primary health care clinics in Western Cape, South Africa | 1,340 including 801 PLWH and 632 patients with Type I/II diabetes: Dedicated Group: 457 including 270 with HIV; Designated group: 438 including 243 with HIV; TAU group: 445 including 288 with HIV | CHWs | Depression severity (CES-D) | Compared with treatment as usual, the dedicated group had statistically significant lower depression severity at six months and at 12 months with a small increase in effect size over time; the designated group had statistically significant lower depression severity at six months and at 12 months with a large increase in effect size over time |
| (Nakimuli-Mpungu et al., 2020) | Pragmatic two-arm cRCT: group support psychotherapy (GSP) vs. group HIV education (active control) | GSP sessions to educate participants on depression symptoms, treatment options, comorbidity of depression and HIV, problem-solving, positive coping, and income-generation skills | 30 primary health centers offering HIV care in three districts (Gulu, Kitgum, and Pader) in post-conflict, rural, northern Uganda | 1,473 PLWH (19 +) with mild to moderate major depression on ART and antidepressant-naïve | CHWs | Major depression (MINI), PTSD symptoms (locally adapted Harvard Trauma Questionnaire), and suicide risk/attempts | Two (<1%) of participants in intervention group vs. 160 (28%) in control group were diagnosed with major depression six months post-treatment and three (1%) vs. 225 (40%) at 12 months; two (<1%) of participants in the intervention group vs. 114 (20%) in the control group reported post-traumatic stress symptoms at six months and one(<1%) vs. 225 (40%) at 12 months; group differences were significant; 25 suicide attempts resulted in four suicide deaths |
| (Nyamayaro et al., 2020) | Case series | Six-session intervention based on PST for depression and barriers to adherence (PST-AD) with stepped care for those with depression whose depression did not recover with PST-AD | Parirenyatwa General Hospital ART clinic in central Harare, Zimbabwe | Nine PLWH (18 +) on ART for at least four months with probable depression and adherence problems (only three were presented in the article) | Adherence counselor | Depression (Self-Reporting Questionnaire–8: SRQ–8) | SRQ–8 scores decreased from baseline to follow-up for all participants |
| (Petersen et al., 2014) | Pilot RCT | Adaptation of a local group-based IPT intervention (8 weekly sessions) | Public clinic with dedicated ART clinic outside Durban, KwaZulu-Natal, South Africa | 76 PLWH with MDD (intervention: 41 and control: 35) | Lay HIV counselors | Depression (PHQ–9 and Hopkins Symptom Checklist–25: HSCL–25)**** | Significantly greater improvement in depression scores on the PHQ–9 in the intervention group; significant decline in the mean scores on the HSCL–25 was found for both groups, although this was more pronounced for the intervention group |

(*Continued*)

| Citation | Study design | Intervention description | Location/setting | Sample size/eligibility | Lay health workers | Mental health outcome(s) | Results |
|---|---|---|---|---|---|---|---|
| (Pokhrel et al., 2018) | Community-based, non-randomized study with intervention and control group (usual care, which includes peer support groups) | Monthly community home-based care and support services with basic health care and ART adherence support and counseling | Community, home-based care in Nepal | 682 (intervention: 344 and control: 338) | CHW, peer, and social worker | Depression (CES-D), anxiety (Composite International Diagnostic Interview Short-Form), and stress (Cohen's Perceived Stress Scale) | Significantly reduced depressive symptoms, anxiety, and stress at six months follow-up |
| (Prinsloo et al., 2016) | Single case pre-test and post-test | Five-month intervention involving the creation of community hubs, three-hour workshops, fruit and education at the clinic, door-to-door education activities, home visitation for support, psychodrama group performances on HIV stigma reduction, and anti-stigma campaigns | Clinic and community "hubs" in Tlokwe, North West Province, South Africa | 62 PLWH and 570 community members | Peers and other LHWs (people not living with HIV with close relationships with PLWH) | Depression (PHQ–9) and well-being (Mental Health Continuum-Short Form) | Average depression score decreased but was not statistically or clinically significant; the intervention had statistically and clinically significant effects on emotional and social well-being but not on psychological well-being |
| (Ross et al., 2013) | Mixed methods RCT (quantitative) with qualitative data collection embedded in experimental design (used to compare with the quantitative findings) | Weekly telephone-based emotional and informational support, as well as prenatal care and information related to pregnancy and self-care | Prenatal clinic in Thailand | 40 PLWH (20 intervention and 20 control) | Nurses | Depressive symptoms (CES-D) | Mean scores for depressive symptoms among participants in the intervention group with telephone support decreased significantly from baseline and one month, as well as baseline and two months, while the symptoms among the control group did not significantly change over time; qualitative interviews suggest that telephone support helped lessen depressive symptoms by putting things in perspective, making the participant feel supported, and alleviating participant concerns through practical guidance |
| (Thomson et al., 2014) | Prospective cohort | Community-based accompaniment to ART delivery: daily home visits to participants to provide social support, monitor for drug side effects and adverse events, identify barriers to adherence, and observe ingestion of medications | District hospital and satellite clinics in Rwanda: five health centers in Kayonza and Kirehe districts where the community-based accompaniment model was added to the national model and four health centers in Musanze district that implemented the clinic-based national model alone | 610 PLWH (304 in intervention group and 306 in control group) | CHWs | Perceived mental health QoL (Medical Outcomes Survey HIV Scale) and depression (HSCL–15)***** | During the first year of treatment, improvements in depression and mental health quality of life were observed with greater effect among people receiving the intervention; after 12 months of clinic-based ART and community-based accompaniment, the average prevalence of depression fell significantly by an additional 44.3% and perceived mental and physical health quality of life significantly improved by more than twice as much as those receiving usual care |
| (Verhey et al., 2020) | Qualitative | The Friendship Bench: CBT-based PST with behavioral activation component aimed at reducing symptoms of common mental disorders | Largest primary care clinic in Mbare, Harare, Zimbabwe | Ten clients living with HIV with trauma histories who met PTSD | CHWs | Common mental disorders (i.e., depression and anxiety), including *kufungisisa kwe njodzi* (excessive thinking due to | LHWs were perceived as helpful in dealing with *njodzi* (trauma); clients described various examples of *njodzi* that stemmed partly from their childhood and partly from current socio-economic and structural circumstances; common sources of |

**Table 1.** (*Continued*)

| Citation | Study design | Intervention description | Location/setting | Sample size/ eligibility | Lay health workers | Mental health outcome(s) | Results |
|---|---|---|---|---|---|---|---|
| | | | | criteria and five LHWs | | trauma – a local equivalent to PTSD) | *njodzi* from childhood included poverty, loss of caretakers/family members, abuse, chronic illness, deprivation, and a struggle for survival; LHWs addressed suicidal ideation and found it common in PLWH with a history of *njodzi* |
| (Wagner et al., 2016) | cRCT | Routine depression screening (PHQ–2) with two task shifting models of post-screening evaluation and treatment: structured protocol model (care was provided largely by trained nurses who acted as depression care managers) or clinical acumen model (i.e., primary care provider determination to further evaluate and treat patients for positive depression screens) | Hospitals operating an HIV clinic on certain days of the week, Uganda | 1,252 (640 at structured protocol clinics and 612 at clinical acumen clinics) | Clinical/ medical officers, nurses, and peers | Screening (PHQ–2) and diagnosis (PHQ–9) of depression and major depression (MINI), as well as prescription of anti-depressants | Participants in the structured protocol arm were significantly more likely to have received further evaluation by a medical provider using the PHQ–9 (84% vs. 49%); among the clinically depressed clients, the advantage of the structured protocol model over clinical acumen was not statistically significant regarding PHQ–9 depression evaluation |
| (Williams et al., 2014) | RCT | Ai Sheng Nuo (i.e., Love, Life, Hope): Home-based intervention for ART adherence with in-person visits (at home, in public venues, or over the phone) from a nurse and peer educator over 6 months | Clinic- and home-based care in Hunan, China | 110 PLWH (intervention: 55 and control: 55) | Nurses and peers | Depressive symptoms (CES-D) | Significant difference in overall depression scores between the two groups with the control group having a higher proportion of people with a CES-D score ≥ 16; comparing baseline CES-D scores to 12-month CES-D scores, the intervention group showed a significant decrease in depressive symptoms compared to the control group; comparing 6-month CES-D scores to 12-month CES-D scores, change in scores did not significantly differ between the two groups, though depressive symptoms increased in control subjects and decreased in intervention subjects |
| (Yu et al., 2014) | Single-arm open evaluative study using a pre- and post-intervention design | Eight 2.5-hour bi-weekly group sessions (10–13 PLWH and six people not living with HIV in each group) over four months based on the resilience framework to improve self-worth, emotional control, optimism, social support, and empathy toward vulnerable people | Rural county in central China | 75 PLWH and 36 people not living with HIV (fellow villagers) in rural China | Family planning officers | Depression, anxiety, and stress (Depression, Anxiety, and Stress Scale); resilience (Connor-Davidson Resilience Scale); and QoL (Medical Outcomes Study HIV Health Survey and WHOQOL-BREF)*** | Resilience and mental health QoL improved significantly and depression, anxiety, and stress decreased significantly at the completion of the intervention; effects were sustained after three months, though not with statistical significance for depression (p = .05) |

*Note*: Some mental health-related outcomes were assessed but not included in this table, such as functioning impairment (*), functioning (**), social support (***), perceived social support (****), and perceived functional social support (*****).

et al., 2022 compared two CHW approaches – dedicated care (representing 100% effort of the CHW) versus designated care (additional responsibilities above and beyond the CHW's usual role) – to treatment as usual.

### Study aims

Studies primarily aimed to assess intervention efficacy or effectiveness (*n* = 21) from pre-post and over time. Twelve (42.9%) of the studies were pilots, some of which also evaluated the acceptability (*n* = 7) and feasibility (*n* = 8) of interventions, e.g., attendance, retention, fidelity, and identifying barriers and facilitators. Stated aims also included exploring the associations between intervention-related independent variables and mental health-related dependent variables to identify predictors, as well as the effects of other variables (e.g., mediation and moderation) on intervention effects. One qualitative study, Verhey et al., 2020, explored the perspectives and experiences of LHWs delivering the intervention and clients receiving it. Wagner et al., 2016 compared the implementation of two depression care intervention models for their advantages and disadvantages in outcomes.

### Location

Based on the 2024 fiscal year World Bank income classification (WB, 2024b, 2024c), which is determined by the 2022 calendar year (the publication year of the most recent study in this review) gross national income per capita as calculated using the World Bank Atlas method (WB, 2024a), nearly all studies were conducted in LMICs (*n* = 27, 96.4%) with over half occurring in the WHO African region (*n* = 16, 57.1%). The U.S. was the only HIC with a study eligible for inclusion in this review. None of the studies in this review were conducted in the European and Eastern Mediterranean regions. China had the greatest representation among all the countries involved in research on this subject matter (*n* = 6, 21.4%). The number of studies in each location is reported by country, WHO region, and World Bank income classification in Table 2.

### Setting

The majority of studies took place at clinics/health centers (*n* = 21, 75%), which were often affiliated with and/or government-funded/operated (*n* = 10, 35.7%), hospitals (*n* = 7, 25%), NGOs/nonprofits

(*n* = 5, 17.9%), and universities/research institutes (*n* = 3, 10.7%). Study sites provided primary care (*n* = 7, 25%) and/or specialized care, e.g., HIV (*n* = 13, 46.4%), infectious diseases (*n* = 2, 7.1%), and prenatal (*n* = 1, 3.6%) care.

### Study sample

The sample size average was 377 (median: 100, range: 9–1,473). Most studies included both men and women. To participate, some individuals were required to have symptoms of mental illness or diagnoses of major depressive disorder (MDD) or PTSD. Other criteria included ART enrollment or the length of time since diagnosis. Some studies included participants who were not PLWH, such as family members (Chen et al., 2018), community members (Prinsloo et al., 2016), diabetes patients (Myers et al. 2020), and other non-PLWH (Yu et al., 2014; Verhey et al., 2020). However, their involvement was typically a key component of intervention design and implementation for the benefit of PLWH (e.g., stigma reduction efforts), except for the study by Myers et al. 2020, and their outcomes were reported separately. Only one study, Verhey et al., 2020, reported studying PLWH with histories of trauma.

### Interventions

Studies assessed novel intervention packages, as well as evidence-based MHPSS interventions and treatment models/frameworks, including problem-solving therapy (PST; *n* = 5), cognitive behavioral therapy (CBT; *n* = 4), interpersonal therapy (IPT; *n* = 3), stepped care model (*n* = 2), self- and family-based management intervention (SAFMI; *n* = 1), and measurement-based care (MBC; *n* = 1). Some studies (Masquillier et al., 2014; Thomson et al., 2014; Williams et al., 2014 and Pokhrel et al., 2018) assessed the impact of adherence support interventions for ART on mental health.

With few exceptions, studies did not disclose whether trauma informed intervention design nor selection. Verhey et al., 2020 assessed the impact of The Friendship Bench (an intervention for common mental disorders and not "trauma-specific") on PLWH with trauma histories in Zimbabwe and explored sources of their trauma. Similarly, Meffert et al., 2021 and Nakimuli-Mpungu et al., 2020 assessed interventions' transdiagnostic potential for treating PTSD.

**Table 2.** Studies by WHO region and World Bank income classification (N = 28)

| WHO region | Low income | Lower middle income | Upper middle income | High income | Total |
|---|---|---|---|---|---|
| African | Uganda (*n* = 2) Ethiopia (*n* = 1) Rwanda (*n* = 1) | Zimbabwe (*n* = 4) Cameroon (*n* = 1)* Tanzania (*n* = 1) Kenya (*n* = 1) | South Africa (*n* = 5) | N/A | 16 |
| The Americas | N/A | N/A | Guatemala (*n* = 1) | United States (*n* = 1) | 2 |
| South-East Asia | N/A | India (*n* = 1) Nepal (*n* = 1) | Indonesia (*n* = 1) Thailand (*n* = 1) | N/A | 4 |
| European | N/A | N/A | N/A | N/A | 0 |
| Eastern Mediterranean | N/A | N/A | N/A | N/A | 0 |
| Western Pacific | N/A | N/A | China (*n* = 6) | N/A | 6 |
| **Total** | 4 | 9 | 14 | 1 | 28 |

*Note*: *Two papers reported findings from the same study. Regional categories are from the World Health Organization (WHO, 2024a) and income-level country classifications are based on the World Bank Fiscal Year 2024 (Calendar Year 2022) – the publication year of the most recent study in this review (WB, 2024b, 2024c).

### LHW characteristics

Interventions mobilized non-specialist HCWs ($n = 14$; e.g., nurses, social workers, adherence counselors, and case managers), peers ($n = 12$; i.e., fellow PLWH), CHWs ($n = 10$; i.e., local community members with clinical roles and responsibilities), and other LHWs ($n = 2$; i.e., traditional medicine practitioners or family/friends/ colleagues of PLWH) to deliver MHPSS services to PLWH. Summaries detailing characteristics of LHW mobilization (i.e., roles, responsibilities, and training) by worker type can be found in Table 3. Few studies reported specific qualifications/eligibility requirements but most disclosed characteristics of LHWs that could factor into selection (e.g., profession/clinic experience, education/literacy, and/or lived experience). However, LHWs also appeared to be recruited for convenience and proximity to service locations.

### Compensation

Compensation for LHWs was rarely disclosed and usually discussed in general terms. Some LHWs were employed (Garfin et al., 2019; Verhey et al., 2020), paid (Thomson et al. 2014), or had funding (Petersen et al., 2014). Nakimuli-Mpungu et al., 2020 estimated the value of voluntary time that LHWs spent facilitating group support psychotherapy (GSP) or group HIV education sessions as $21.62 USD or the equivalent of three full-time equivalent (FTE) days based on their 2017 earning potential in Uganda ($199 USD per month or about $7 USD per FTE day), which they deemed very cost-effective.

### Intervention duration and delivery

The duration of the interventions ranged from 1 to 18 months. Interaction between LHWs and participants ranged from daily to once per month. However, contact occurred on a weekly basis in half of the interventions studied. Interventions were mostly conducted in person, although some took place over the phone or the Internet.

### Mental health outcomes

Depression and related outcomes (e.g., major depressive disorder, depressive symptoms, and depression severity) were assessed by nearly 90% of studies, which more often used either the Patient Health Questionnaire-9 (PHQ-9; and abbreviated versions) and/or the Center for Epidemiological Studies Depression Scale (CES-D). Other mental health conditions assessed were anxiety ($n = 7$, 25%) and PTSD/trauma ($n = 3$, 10.7%). Four (14.3%) of the studies measured or identified suicide risk, ideation, attempts, and/or deaths during the process of data collection. Quality of life ($n = 5$, 17.9%), social support ($n = 5$, 17.9%), and other mental health-related outcomes (e.g., hope, cognitive escape, stress, resilience, and well-being) were also assessed. Three studies assessed the frequency of activities along the mental health continuum, such as screening, evaluation and diagnosis, treatment (i.e., prescription), referral, and navigator interactions.

### Synthesis of results

Most studies reported statistically significant improvements in mental health outcomes from baseline to follow-up and between groups post-intervention. Conversely, Guy et al., 2022 found that a peer support intervention to improve treatment engagement among African Americans living with HIV and serious mental illness was successful in rendering the mediation effects of internalized stigma on the association between HIV discrimination and cognitive escape coping insignificant.

Usually, the statistical significance of positive effects was sustained when measured over time (up to 12 months). However, Yu et al., 2014 observed that although the effects on depression were sustained, they were no longer significant after 3 months ($p = .05$). The authors posit that this change could be explained by the study's relatively small sample size and that the small to moderate effect size (.29) at follow-up may become statistically significant with a larger study sample. Conversely, Masquillier et al., 2014 found that peer adherence support did not have a direct effect on the level of hope among PLWH but that better family function significantly increased the positive effect of the intervention at the second follow-up (2–3 years post-intervention).

Other studies found that significance differed based on the outcome being assessed (Yu et al., 2014; Prinsloo et al., 2016; Han et al., 2020; Davis et al., 2021; Li et al., 2021) or on the type of support, participants received (Davis et al., 2021). Abas et al. (2018) found that while participants had lower mean depression scores on the PHQ-9 and the Shona Symptom Questionnaire-14 (SSQ-14), the effects were only significant on the PHQ-9. This could be explained by differences in sensitivity, specificity, and cultural relevancy of the instruments used or that the SSQ-14 is a scale for common mental disorders (e.g., anxiety) and not specific to depression.

Studies did not report any adverse effects due to intervention participation, however, Fajriyah et al. (2018) found that other variables (i.e., stigma and depression) negatively influenced the quality of life. A few studies did not assess the statistical significance of their findings. Duffy et al. (2017) provided only descriptive statistics (e.g., frequency of positive screens), Nyamayaro et al. (2020) used case studies to illustrate findings from a case series, and Verhey et al. (2020) was a qualitative study.

### Implementation barriers/facilitators

The most common challenges were the burdensome amount of time and effort required to implement the intervention ($n = 9$, 32.1%; e.g., recruitment, training, supervision, and support), cost of implementation/participation ($n = 7$, 25%), and distance from study site/access to transportation among participants ($n = 5$, 17.9%). Other challenges faced by participants included concerns about the intervention (e.g., lack of trust/comfort or privacy and confidentiality), literacy issues (e.g., inability to complete assignments or lack of knowledge/understanding of mental health), and poor help-seeking (e.g., fear of being stigmatized, unfamiliarity with the intervention and benefits, scheduling challenges/competing priorities, and willingness to participate).

Clinic-level barriers were mostly structural: infrastructure/capacity (e.g., lack of meeting space, routine data collection/health records system, or trained personnel), health system incompatibility (e.g., lack of collaboration among different service providers or ability to integrate services), and lack of available resources (e.g., availability of psychosocial services/alternative treatment and ability to address clients' social service needs).

Finally, some studies ($n = 9$, 32.1%) identified issues with fidelity to the intervention, including compliance with intervention protocol, attendance/retention among participants, and LHW inability to deliver services due to insufficient training or guidance.

**Table 3.** Characteristics of LHW mobilization by worker type

| LHW | Studies | Roles | Responsibilities | Training description/key components |
|---|---|---|---|---|
| Non-specialist HCWs (e.g., nurses, social workers, adherence counselors, and case managers) | (Ross et al., 2013; Pence et al., 2014; Petersen et al., 2014; Williams et al., 2014; Yu et al., 2014; Gaynes et al., 2015; Wagner et al., 2016; Duffy et al., 2017; Li et al., 2017; Andersen et al., 2018; Chen et al., 2018; Pokhrel et al., 2018; Han et al., 2020; Nyamayaro et al., 2020; Asrat et al., 2021) | • Interventionist<br>• Counselor/therapist (e.g., CBT/CBT-AD, IPT, and PST)<br>• Depression care manager<br>• Supervisor of peer therapists<br>• Home-based care support team member | • Delivering, implementing, or supervising the delivery of individual and group therapeutic interventions (e.g., CBT/CBT-AD, PST, SAFMI, group IPT, stepped care approach, MBC, relaxation training, basic counseling/micro-counseling, and family and spousal counseling)<br>• Screening/evaluating, diagnosing, treating, and making referrals<br>• Psycho-education and problem management<br>• Home visits to provide adherence support, education, and comprehensive care<br>• Comprehensive social, emotional, and informational support<br>• Data collection<br>• Developing protocol for mental health emergencies (i.e., suicidal ideation)<br>• Facilitating intervention/group activities: building, handcrafts, recreational games, and volunteer work | • Didactic and interactive (e.g., roleplaying) training<br>• Most trainings took place over the course of one to seven days, although one intervention called for one year of training to achieve Level II certification as a national psychological counselor<br>• Example training topics:<br>○ How to implement or train others to implement the intervention (e.g., WHO group IPT, MBC, and stepped care) and study protocol (e.g., referral procedures and data collection responsibilities)<br>○ Diagnosing and managing symptoms (e.g., identification of major depression, medication adherence, and delivery of medication treatment recommendations), an integrated approach to mental health, and psycho-social support/counseling/educational skills (e.g., therapeutic communication and Freirian educational techniques)<br>○ Other relevant subjects: HIV/AIDS, sexuality, substance abuse, and HIV care<br>• Supervision: weekly or monthly by study investigators, psychiatrists, therapists, and/or clinical psychology trainees for ongoing feedback and quality assurance (e.g., direct observation and reviewing narrative logs)<br>• Assessment: Although not a common practice, several interventions required trainees to pass a final written/competency test (must score >70%–80%), daily evaluations or fidelity checks after each session, and/or theoretical and practical examinations |
| CHWs | (Thomson et al., 2014; Duffy et al., 2017; Pokhrel et al., 2018; Garfin et al., 2019; Nakimuli-Mpungu et al., 2020; Verhey et al., 2020; Davis et al., 2021; Meffert et al., 2021; Kaaya et al., 2022; Myers et al., 2022) | • Interventionist and supervisor<br>• Therapist (e.g., group psychotherapy, IPT, and PST)<br>• Health navigator<br>• Community behavioral health worker (CBHW)<br>• *Ambuya utano* (grandmother health provider)<br>• ASHA (accredited social work activist)<br>• Home-based care support team member | • Conducting individual and group counseling/therapy sessions (e.g., CBT, PST, GSP, individual/family/spousal counseling, and basic counseling) or home visits<br>• Screening and making referrals<br>• Health education and promotion (e.g., health campaigns and vaccination drives)<br>• Reminders for and accompaniment to clinic appointments/group sessions<br>• Adherence support (e.g., direct observation, nutrition support, socioeconomic support, and transportation)<br>• Social and emotional support (e.g., assistance with disclosure of HIV status, motivational messaging, showing empathy and loyalty, discussing life challenges, and promoting self-care) | • Didactic and interactive (e.g., roleplaying and practice cases/sessions, brainstorming, and small group discussions) training. A training-of-trainers model was used by some interventions to expand the reach of training to other LHWs<br>• Duration varied (from two days to four months)<br>• Example training topics:<br>○ Intervention (e.g., CBT, IPT, PST, formal and informal GSP by Makerere University/Ministry of Health, WHO mhGAP depression treatment, basic counseling and skills building, motivational interviewing, home-based palliative care and psychosocial support, and Western Cape Department of Health CHW approach) and study protocols (e.g., referrals) |

*Cambridge Prisms: Global Mental Health*

**Table 3.** (*Continued*)

| LHW | Studies | Roles | Responsibilities | Training description/key components |
|---|---|---|---|---|
| | | | • Maintaining regular communication/check-ins<br>• Patient advocacy and addressing barriers to accessing care<br>• Creating a protocol for mental health emergencies (i.e., suicidal ideation) and managing distressed participants and risk of harm<br>• Resource management | ○ Screening, diagnosing, and managing mental health conditions (e.g., depression); basic therapeutic counseling and psychosocial support skills; and follow-up care (e.g., risk management, adherence support, social and emotional support, sleep hygiene, and structured physical activity)<br>○ Other relevant subjects: HIV (e.g., disease progression, PLWH needs, confidentiality, ART adherence, coping and self-care strategies, livelihood skills, and comprehensive support), alcohol and substance use, resource mobilization, peer supervision, and advocacy and leadership skills (e.g., supervision, problem-solving, and trust-building)<br>• Supervision: Sometimes weekly individual or group observation/supervision of activities (e.g., practice cases) either in-person or via phone by trained CBHWs, local psychiatrists, psychiatrist experts, on-site peer supervisors, and previously trained HCWs to assess proficiency<br>• Assessment: A few studies reported conducting assessments post-training and periodically (e.g., quarterly for quality assurance). One study reported evaluating protocol adherence for fidelity and quality on a scale of 1–10 (must score ≥9 to pass and maintain consistent high scores for selection; one to two additional practice cases needed for those who fail) |
| Peers | (Masquillier et al., 2014; Williams et al., 2014; Prinsloo et al., 2016; Wagner et al., 2016; Li et al., 2017; Abas et al., 2018; Fajriyah et al., 2018; Pokhrel et al., 2018; Asrat et al., 2021; Davis et al., 2021; Li et al., 2021; Guy et al., 2022) | • Interventionist<br>• Therapist (e.g., group IPT and PST)<br>• Mental health screener<br>• Health navigator<br>• Peer adherence supporter<br>• Peer support group participant<br>• Home-based care support team member<br>• Community mobilizer<br>• Recruiter ("field worker") | • Delivering PST, individual/family/spousal/peer counseling, and group intervention sessions<br>• Screenings/assessments and referrals<br>• Conducting home visits<br>• Sending reminders for and providing accompaniment to clinic appointments<br>• Health education and promotion, including workshops to the community and psychodrama group performances about HIV stigma reduction and an HIV stigma-reduction community project involving home visits to PLWH<br>• Adherence support (e.g., side effects, stigma, and other socioeconomic factors)<br>• Social and emotional support (e.g., assistance with disclosure of HIV status, motivational messaging, discussing life challenges, and building confidence)<br>• Nutrition support<br>• Maintaining regular communication/check-ins | • Didactic and interactive (e.g., brainstorming, brief presentations, group discussion, and roleplaying) training. Peers sometimes had prior training/background knowledge from clinic work experience, lived experience (HIV and ART), secondary school education, and/or training in HIV/AIDS basic counseling, HIV stigma and coping skills, and/or skills to plan and implement an HIV stigma-reduction community project<br>• Duration varied (from four to seven days)<br>• Example training topics:<br>○ Intervention (e.g., WHO group IPT, home-based palliative care, and peer adherence support) and study protocols (e.g., data collection and record-keeping)<br>○ Psychosocial support and support group leadership<br>○ Other relevant subjects: HIV (e.g., ART and medical adherence, sexuality, stigma and coping, nutrition, and infection control in the |

(*Continued*)

| LHW | Studies | Roles | Responsibilities | Training description/key components |
|-----|---------|-------|------------------|-------------------------------------|
| | | | • Patient advocacy and addressing barriers to accessing care<br>• Recruitment and enrollment<br>• Resource management<br>• Record keeping | home), substance use, Freirian educational techniques, and community mobilization<br>• Supervision: Rarely mentioned but a few interventions reported methods, including weekly meetings to discuss caseload, periodic review (some variability around provision and documentation of support), and detailed narrative logs of content and process of home visits. Supervision was given by a psychologist or research team members<br>• Assessment: Rarely mentioned but one intervention (i.e., WHO group IPT) required daily assessments and an exit exam (must score ≥ 70% to pass) |
| Other natural helpers (e.g., traditional medicine practitioners, and family/friends/colleagues of PLWH) | (Prinsloo et al., 2016; Duffy et al., 2017) | • Interventionist/therapist<br>• Community mobilizer | • Providing basic counseling and therapeutic interventions<br>• Screenings and referrals<br>• Creating a protocol for mental health emergencies (i.e., suicidal ideation)<br>• Health education and promotion (e.g., workshops for the community, psychodrama group performances about HIV stigma reduction, and an HIV stigma-reduction community project involving home visits to PLWH)<br>• Nutrition support | • Didactic and interactive training (e.g., lectures, discussions, and roleplaying). Prior HIV-related knowledge/training was required by Prinsloo et al., 2016 (i.e., knowledge about understanding and coping with HIV stigma, as well as skills to plan and implement their own HIV stigma-reduction community project)<br>• Duration: Duffy et al., 2017 implemented a two-day training of trainers in two phases, whereas, Prinsloo et al., 2016 offered an initial four-day workshop and an additional two-day workshop for some participants to be trained as fieldworkers in data collection<br>• Example training topics:<br>○ How to implement or train others to implement the intervention (i.e., stepped care model or community hubs)<br>○ Study protocol (e.g., referral protocol and tools, data collection, and record-keeping)<br>○ Mental health disorders, the stepped-care mental health and HIV integrated approach, and therapeutic communication<br>○ Other relevant subjects: alcohol and substance use and leadership skills |

*Note:* This is a summary of what was reported across studies but is not representative of every study.

However, Anderson et al. (2019) suggested that continued education (e.g., booster training to review session content) and implementation of "peer supervision models" (e.g., from fellow nurses trained to implement CBT) may help improve fidelity to the intervention. Additionally, Verhey et al. (2020) emphasized the need to provide LHWs with clear instructions for delivering MHPSS services to PLWH in need of support (e.g., how to address symptoms of PTSD in PLWH once they are recognized), while Petersen et al. (2014) highlighted the need to provide organizational leaders with a human resource plan (e.g., LHW role, the scope of practice, and supervision structure) and education about the value of interventions being implemented and service requirements.

Studies also reported a variety of measures taken to facilitate implementation. Researchers and LHWs worked to establish trust through relationship building (e.g., care teams visiting participants at home/off-site or by giving participants continuity with the same service provider) and to ensure participants felt supported. Compensation, transportation, and/or food assistance were provided to help defray the costs of participating in interventions, which were often tailored to the target population (e.g., culturally appropriate, community-informed, client-driven/patient-centered, appropriate education/knowledge level, and delivered in local language) and easy for lay people to deliver. Intervention acceptability was reported as a key component of success. A couple of studies also reported using fidelity checks to ensure interventions were being delivered as intended.

### Gaps/future research

Studies in this review self-reported several limitations, mainly related to study design. Low sample size was common (many studies reviewed were pilots), so findings were often preliminary and underpowered. Some studies did not have comparison groups or did not use blinding nor randomization techniques when they did. Limitations of data instruments and omissions in data collection/analysis were also reported. A few studies reported potential contamination or confounding.

To address these limitations, a common suggestion for future research was to pursue large-scale efficacy and effectiveness studies (i.e., RCTs and mixed methods) with control/comparison groups. Researchers also proposed new lines of inquiry and adjustments/changes to interventions (e.g., fewer sessions, using a train-the-trainer approach, and other modalities), as well as alternative methods of data collection (e.g., type of data and measures used) and analysis (e.g., mediation/moderation and subgroup analyses). To better understand the full potential of interventions, several studies recommended assessing effects over longer periods and assessing HIV outcomes. It was also suggested that studies try integrating mental health services (e.g., screening and treatment) into HIV/chronic disease service delivery. Studies called for more research on depression and other common mental disorders, alcohol and substance use, and PTSD.

### Discussion

Findings indicate that MHPSS task shifting and task sharing intervention studies published from 2013 through 2022 yielded promising results for PLWH. Studies primarily aimed to assess the efficacy and effectiveness of interventions, and nearly all mental health outcomes measured significantly improved. In our sample, the distribution of studies involving non-specialist HCWs, CHWs,

and peers was relatively even, and there was no clear indication nor study showing that one type of LHW should be preferred over the other in terms of efficacy/effectiveness. It is likely that service delivery model selection may solely depend on the priority given by implementers to clinic role/experience, lived experience, and community knowledge/standing when delivering MHPSS services based on context and setting. In fact, this review suggests that significant improvements in the mental health outcomes of PLWH can be achieved when MHPSS services are delivered by LHWs of any kind.

This scoping review identified several gaps in the literature. First, as predicted, there was a great divide between LMICs and HICs concerning the production of literature on this research subject. Within LMICs, as income classification increased, so did the number of studies. HICs are the exception with the least number of studies. Although we had hoped to gain more insight into interventions being used in HICs, only one study was eligible for inclusion in our review, highlighting the continued need for more studies to be conducted in these settings to facilitate the bi-directional translation of knowledge and exchange of ideas between LMICs and HICs. This is of particular importance in HICs with a large mental health provider gap, such as in the U.S., where there is a confluence of a national mental health professional shortage (RHIhub, 2025) and high unmet needs (an estimated 46.2% of adults with mental illness do not receive treatment) (SAMHSA, 2024). Mental health providers may be unable to respond to high patient volume at HIV clinics, thus exacerbating problems with access to and utilization of mental health services, which may account for a significant proportion of PLWH with needs not receiving them (an estimated 27%) (Reif et al., 2006; CDC, 2024).

This review adds to growing evidence suggesting that task shifting and sharing interventions are valuable tools in resource-limited settings. The lack of studies in HICs may be explained by barriers such as licensing or certification to deliver certain therapies, as well as the associated costs and pre-requisites (e.g., higher education, prior training, and specialization); it may be time to reconsider what qualifications are necessary to deliver MHPSS services to PLWH and to establish pathways empowering LHWs to attain them.

Second, interventions specifically for PTSD among PLWH (Nakimuli-Mpungu et al., 2020; Verhey et al., 2020; Meffert et al., 2021) composed only a small fraction of studies in this review, and trauma did not otherwise appear to factor intentionally into the design nor selection of interventions in our sample. CBT – a therapy strongly recommended by the American Psychological Association's Clinical Practice Guideline for the Treatment of PTSD (APA, 2017) – was implemented or adapted by several studies in our sample to treat common mental disorders, such as depression and anxiety. However, Verhey et al., 2020, which assessed a CBT-based PST intervention for PLWH with trauma histories, noted that the intervention "was not a trauma-specific intervention that can be equated with recognized PTSD treatments." The study authors also suggested that future studies integrate PTSD screening and management into LHW-delivered care (Verhey et al., 2020).

Our review suggests that there has been little progress in this area since LeGrand et al., 2015 and that there is an urgent need for more studies to assess the efficacy/effectiveness of trauma-informed interventions for this population and whether LHWs can also deliver them. Promising findings from a recent systematic review and meta-analysis conducted by Connolly et al. (2021)

indicate that professionally trained lay counselors in LMICs were associated with a significant, medium-sized improvement in mental health symptoms (where PTSD was a primary outcome in most interventions studied) across populations and settings.

Third, several studies in this review assessed suicidality among PLWH and identified concerning levels of suicidal ideation, risk/attempts, and deaths. Notably, Duffy et al. (2017) reported that 61% of PLWH that screened positive for depression and/or anxiety also experienced suicidal ideation, Verhey et al. (2020) found that suicidal ideation was common among PLWH with a history of *njodzi* (trauma), and Nakimuli-Mpungu et al. (2020) observed 25 suicide attempts over a 2-year period following an 8-week intervention for PLWH with depression, including four suicide deaths (two participants in GSP and two participants in the active control group within 6 months post-intervention). These findings align with a recent study by the Penn Center for AIDS Research, which found that PTSD and MDD were predictive of suicide risk (Brown et al., 2020), and a systematic review of 43 studies across all six WHO regions from 2010 to 2021, which identified a high risk of suicidality among PLWH (Tsai et al., 2022).

Within our sample, MBC administered by a non-specialist nurse depression care manager reduced suicidal ideation over a 12-week intervention period (Pence et al., 2014; Gaynes et al., 2015) and fewer participants reported suicide risk in GSP delivered by trained LHWs than in an active control group (Nakimuli-Mpungu et al., 2020). Identifying task shifting and sharing intervention strategies that are effective in improving suicide-related outcomes is an important area for future study.

Fourth, recruitment/selection, compensation, supervision, and assessment of LHWs were rarely reported in detail or at all, making it difficult to identify common practices. Important questions remain, such as: (1) if there is any association between the qualifications of LHWs and their ability to deliver services, (2) how compensation affects turnover/quality of care/availability/reliability, (3) if supervision is associated with greater fidelity to intervention protocol, and (4) if training is sufficient. Understanding what these components are in practice and their impact on the success of MHPSS interventions delivered by LHWs is critical to facilitating the translation of this research into broader use.

Finally, budgets and costs associated with implementation beyond participant compensation were rarely disclosed, if at all. Cost-effectiveness was only assessed by Nakimuli-Mpungu et al. (2020). We recommend collecting cost data and conducting this type of analysis in future studies, when possible, as it is an essential consideration for implementation in resource-limited settings.

To our knowledge, this scoping review is the first to focus on MHPSS task shifting and sharing interventions for PLWH globally and inclusive of both LMICs and HICs, which is important for facilitating the translation of knowledge and practices used in different contexts and settings. However, our review had several limitations. We only reviewed English-language studies; thus, important work being done in non-English speaking countries would not have been part of our search returns if published in another language. We omitted acceptability and feasibility papers that described interventions aimed at improving mental health among PLWH but did not report on mental health outcomes for the intended recipients or only reported outcomes based on LHWs' perspectives. The desired outcomes may have been reported elsewhere but those papers did not appear in our search returns and we did not actively search for other papers related to

an intervention nor contact authors to find information about data items that were not reported. Therefore, the interventions described in our review should not be considered the full extent of research on this topic but what was accessible.

To best represent and characterize the available literature on this topic, we also did not exclude studies based on methodological quality nor bias, as these assessments are not typical of scoping reviews, which differ from systematic reviews in their aims (Munn et al., 2018; Tricco et al., 2018). However, we noted these risks as study limitations where applicable. We observed that many studies reviewed were pilots with low sample sizes; there is a need for larger efficacy/effectiveness studies, preferably RCTs and mixed methods. Following intervention participants over longer periods of time could also improve understanding of the long-term mental health benefits of MHPSS services delivered by LHWs.

## Conclusion

There has been growing interest in MHPSS task shifting and task sharing interventions for PLWH in recent years. The findings from this review are promising, however, further research is necessary to improve understanding of the impact of these interventions, such as conducting a systematic review and meta-analysis to estimate effect size, comparing intervention models (i.e., by LHW type and by role in service delivery), testing trauma-informed interventions against trauma-related outcomes, and investigating their potential application in preventing suicide. Gaps in knowledge remain concerning the implementation of MHPSS interventions delivered by LHWs, particularly in recruitment/selection, compensation, and training (e.g., supervision and assessment). Assessing the cost-effectiveness of interventions should be prioritized and may be necessary to persuade HICs to further engage in future research and facilitate the adoption of these service delivery models.

**Open peer review.** To view the open peer review materials for this article, please visit http://doi.org/10.1017/gmh.2025.10013.

**Supplementary material.** The supplementary material for this article can be found at http://doi.org/10.1017/gmh.2025.10013.

**Data availability statement.** The studies included in this review are publicly available online through their publishing journals, although some require an institutional membership or a fee to access them.

**Acknowledgements.** This study was supported by the Woodruff Health Sciences Center Library at Emory University. We thank Sharon Leslie, MS, for her consulting services while conducting the literature search. The content of this article is solely the responsibility of the authors and does not necessarily represent the official views of Emory University. The authors have no conflicts of interest to disclose.

**Author contribution.** The design of this scoping review and its parameters were developed by **C.W.K.** (PI) and **J.M.S.** The literature search was conducted by **C.W.K.** Studies were reviewed and data was extracted by **C.W.K.** and **M.S.C.** with **J.M.S.** serving as a third reviewer. Data analysis was conducted by **C.W.K.** All review processes were guided by **J.M.S.**, **A.S.K.**, **B.A.W-J.**, and **B.G.D.** This article was authored by **C.W.K.** with editorial support from the other authors.

**Financial support.** This review was supported by the Hope2Action Study grant (R01MH121962) to **J.M.S.** and **A.S.K.** from the National Institute of Mental Health of the National Institutes of Health. This work was further supported by the Center for AIDS Research at Emory University (P30 AI050409). The content is solely the responsibility of the authors and does

not necessarily represent the official views of the National Institute of Mental Health or the National Institutes of Health.

**Competing interests.** The authors have no conflicts of interest to declare.

**Ethics.** This study was not subject to ethical review as it did not involve the participation of human subjects.

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
