## [Reviewer Report]

Reviewer comments

Comments to the Author:

Dear Authors,

Thank you for the opportunity to review this global scoping review of task shifting and sharing interventions to improve the mental health of people living with HIV/AIDS. This is an important topic, however, several areas of improvement would strengthen the publication. I have outlined the suggestions below

Introduction

• This section has some epidemiological data; however, this data is not focussed.

The rationale for this global scoping review is not clear.

What is the research question that this review wants to answer?

Paragraphs are too long, consisting of several sentences. Consider paragraphs of 3 - 4 sentences.

Method

This section has some areas for Improvement

Search Strategy Details:

While the search terms are described, their exact combinations (e.g., Boolean operators, truncations, and variations) could be presented more explicitly. Providing a supplementary table with the complete search strategy would enhance reproducibility.

Limiting the review to English-language studies introduces a language bias that may exclude significant research from non-English-speaking regions, particularly in global health contexts.

Mentioning Covidence as the systematic review management platform is helpful, but details about the specific tools or features (e.g., duplicate identification, coding categories) used to manage the review process would provide more transparency.

The definition of task shifting/sharing as involving “non-professionals (i.e., LHWs)” excludes individuals with prior education or training, which could overlook interventions led by trained community health workers who still fit within a task-shifting model in some contexts. Clarifying the rationale behind this exclusion could strengthen the validity of the review.

Inter-Rater Reliability: While discrepancies were resolved through discussion or consultation with a third reviewer, including an explicit measure of inter-rater reliability (e.g., Cohen’s kappa) would strengthen the validity of the study selection process.

More detail on how trauma-informed interventions were identified and categorized could enhance the understanding of contextual factors influencing study inclusion.

RESULTS

This section has some areas for Improvement

While the detail provided is impressive, the narrative is dense and could benefit from a more concise synthesis to avoid overwhelming the reader. Key findings could be summarized with bullet points or tables for easier digestion.

The absence of a quality assessment for included studies is acknowledged as a limitation but remains a notable gap. While this aligns with the purpose of a scoping review, a brief discussion of methodological rigor and potential biases in the included studies would strengthen the review’s interpretability.

While some descriptive statistics are provided (e.g., sample size range, intervention duration), the inclusion of additional quantitative summaries, such as the proportion of studies reporting significant improvements or the frequency of specific implementation barriers, could enhance clarity.

Visual aids, such as charts or tables, would help summarize trends (e.g., geographic distribution, intervention types, outcomes assessed).

The results present a wide array of findings but could better integrate these into a cohesive narrative. For example:

How do specific barriers influence intervention outcomes or study designs?

Are certain types of interventions more feasible or effective in particular settings (e.g., clinics vs. community-based)?

Are there patterns in intervention efficacy based on the type of LHW mobilized?

While most studies reported positive effects, the discussion of cases where effects were short-lived or not significant could be expanded to explore potential reasons or implications.

---

## [Reviewer Report]

Thank you for writing this important global scoping review on task shifting/sharing interventions to improve the mental health of PLWH. I enjoyed reading this paper, and I think it provides a valuable contribution to the literature. You wrote a thorough introduction, including the physical health outcomes impacted by mental health challenges faced by PLWH. Your tables and PRISMA figure were clearly constructed and helped me, as a reader, better understand your findings. I also appreciate your attention to culturally responsive interventions (including your note about njodzi being the local equivalent to PTSD in the Verhey et al 2020 study). I support publication of your manuscript in this journal and suggest the following minor revisions:

1) I recommend updating/adding more recent citations in the introduction where you cite Boarts et al 2009. In this part, I would consider citing relevant, recent HIV stigma literature, especially since you mention stigma in your Results. As you mentioned, stigma plays a big role in physical and mental health outcomes for PLWH.

2) I recommend more recent citations on QoL as well (see Crepaz et al 2008 reference).

3) In the introduction, you mention the extensive training needed to provide CBT in the U.S. I would double-check requirements, as it is common for master’s level clinicians (mental health counselors, social workers, and marriage and family therapists) to provide CBT to clients after they have graduated from their program but before they’ve finished post-master’s clinical supervision requirements for licensure.

4) Line 53, page 8: Please indicate the type of currency for $21.62.

5) Type out the name of the measures listed within the text (separate from the tables). For example, PHQ-9 and CES-D on p.9 “Mental health outcomes”.

6) Line 10, page 10: Please add some examples of how “Researchers and LHWs worked to build trust and ensure participants felt supported”.

7) You found that four studies included suicidal ideation/attempt outcomes in your review. I noticed that Nakimuli Mpungu et al 2020 (p. 27, Table 1) included a sample of 1473 PLWH with n=25 suicide attempts resulting in 4 suicide deaths. Given the nature of PTSD to increase suicidality among individuals experiencing it, please add more information in your Discussion section about implications for future research and how we might address suicidality in task shifting/sharing trauma interventions.

8) Please ensure there aren’t any typos/errors in your tables (for example, proofread Results section of Chen et al 2018 in Table 1).

9) Remember to use non-stigmatizing language throughout (see NIAID HIV Language Guide). Example: Use “people not living with HIV” instead of “non-infected” (p. 35, Table 3).

Thank you!

---

## [Editor Report]

Dear Authors - thank you for the opportunity to review your manuscript, A global scoping review of task shifting and sharing interventions to improve the mental health of people living with HIV/AIDS.

The Reviewers showed enthusiasm for the topic area and approach; however, several methodological concerns were noted. I invite you to respond to their comments.

I also have a few additional questions/comments:

1. The manuscript begins (abstract) by stating that “Due to high levels of trauma, people living with HIV (PLWH) often experience co-morbid/co-occurring mental health conditions...” I propose that this is only partially true, in that many people who acquire HIV do not have trauma histories but still experience mental health conditions at rates higher than people without HIV. It could be that you intend to focus on the (very real) issue of trauma for many people with HIV and the low-intensity interventions for them; however, the search terms for the interventions did not include a trauma variable. I’d argue that this manuscript is really about low-intensity mental health interventions for people with HIV (with and without trauma histories). Whatever the case, I ask that you hone in on the main objective of this paper and, if it is to be centered on trauma-related interventions, to be more explicit about that (and justify the search strategy that does not include trauma as a search variable). 

2. Task sharing and task shifting seem conflated (i.e., when written as task shifting/sharing). These terms relate to similar but different strategies and do not necessarily indicate that a non-professional is involved (although that is often the case). Task shifting occurs when a task is transferred or delegated, while task sharing occurs when tasks are completed collaboratively between providers with different levels of training. Could you please clarify further the strategy or strategies you were most interested in (and/or clarify how these terms are being used in the manuscript with a justification if it is decided to combine them)?

3. What was the rationale for limiting the search to 10 years?

4. I’d like to highlight R1’s comment about quality assessment as a limitation (that you note) that would benefit from greater discussion. 

5. RE “CBT can only be delivered by certified mental health professionals who have obtained at least a master’s degree in a related field, six years of supervised post-graduate experience providing CBT, letters of recommendation, [etc.] ...” - While this may be required for some professional licensees, this is certainly not true within the US or globally. Many mental health professionals use CBT but would not meet these criteria (e.g., in the US, licensed clinical social workers). Moreover, many low-intensity MH interventions include components of CBT, and there are hundreds of books on CBT for self-help.

Best, Jerome

---

## [Reviewer Report]

Thank you for addressing the editor and reviewer comments. Your revision adequately addressed comments regarding varying training and licensure requirements to provide CBT, as well as a clearer description of task sharing and shifting and the studies you included in your scoping review. I appreciate your additional references for QoL, consistent use of non-stigmatizing language, and additional discussion of the implications of suicidality in the reviewed studies, as well as in your abstract and conclusion. Re-framing your paper to note trauma interventions as a secondary interest was helpful. Your scoping review will be a great addition to the literature. I have some additional comments:

1) Please ensure once more that all acronyms are spelled out at first mention in the manuscript text (Group Support Psychotherapy acronym in the Discussion section).

2) Please indicate the amount of time included in the $21.62 USD cost to compensate LWH in the Nakimuli-Mpungu et al 2020 article. Is it per hour or per session?

3) A more recent reference on the number of adults with untreated mental illness in the United States (see Walker et al., 2015 in the Discussion).

4) You mention four future research questions for identifying common practices for recruiting, training, compensating, supervising, and assessing LHWs in the Discussion. After reading that, I returned to the Implementation barriers/facilitators subsection under Results. Did any of the nine studies that cited fidelity issues elaborate on the LHW’s inability to deliver services due to insufficient training or guidance? An additional sentence or two would be helpful, especially as you state in the next paragraph that interventions were often tailored to be “easy for lay people to deliver”.

5) Please proofread the tables again (“QQ” in Li et al 2021 sample size/eligibility in Table 1).

---

## [Reviewer Report]

The authors have comprehensively addressed all of my previous comments, resulting in significant improvements to their manuscript. I believe the manuscript now makes an important and valuable contribution to the existing literature. I therefore strongly recommend that it be accepted for publication.

---

## [Editor Report]

Dear Authors:

Thank you for your comprehensive modifications in response to the Reviewers' comments.

One Reviewer has just a few remaining, minor issues, I hope you’d be willing to address after which I can formally decision this manuscript.

Many thanks, Jerome

---

## [Reviewer Report]

Thank you so much for your thoughtful responses and addressing my previous comments. This is an excellent article that contributes to existing literature.